



# Brief communication: 4 Mm³ collapse of a cirque glacier in the Central Andes of Argentina

Daniel Falaschi[1,2], Andreas Kääb[3], Frank Paul[4], Takeo Tadono[5], Juan Antonio Rivera[2], Luis Lenzano[1,2]

1. Departamento de Geografía, Universidad Nacional de Cuyo, Mendoza, 5500, Argentina
2. Instituto Argentino de Nivología, Glaciología y Ciencias Ambientales, Mendoza, 5500, Argentina
3. Department of Geosciences, University of Oslo, Oslo, 0371, Norway
4. Department of Geography, University of Zürich, Zürich, 8057, Switzerland
5. Earth Observation research Center, Japan Aerospace Exploration Agency, Tsukuba, 305-8505, Japan

*Correspondence to:* Daniel Falaschi (dfalaschi@mendoza-conicet.gov.ar)

**Abstract.** Among glacier instabilities, collapses of large parts of low-angle glaciers are a striking, exceptional phenomenon. So far, merely the 2002 collapse of Kolka Glacier in the Caucasus Mountains and the 2016 twin detachments of the Aru glaciers in western Tibet have been well documented. Here we report on the previously unnoticed collapse of an unnamed cirque glacier in the Central Andes of Argentina in March 2007. Although of much smaller ice volume, the $4.5 \pm 1 \times 10^6$ m³ collapse of Leñas glacier in the Andes is similar to the Caucasus and Tibet ones in that the resulting ice avalanche travelled a total distance of ~2 km over a surprisingly low angle of reach (~5°).

## 1 Introduction

On steep glacier fronts, icefalls, and hanging glaciers (usually >30º steep), glacier instabilities in the form of ice break-offs and avalanches of varying size and magnitude are common and have been noted everywhere around the globe (Faillettaz et al., 2015). The current WGMS '*special events*' database lists in fact a total of 110 ice avalanche events worldwide (WGMS, 2017). Such gravitational ice failures can be a normal process of ablation of steep glaciers, but extraordinary events can be triggered by seismic events, or changes in the ice thermal regime, or in topographic or atmospheric conditions (Faillettaz et al., 2015). Typical volumes of ice avalanches from steep glaciers are in the order of up to several $10^5$ m³, with extraordinary event volumes of up to several $10^6$ m³. Yet the detachment of large portions of low-angle glaciers is a much less frequent process, and has so far only been documented in detail for the $130 \times 10^6$ m³ avalanche released from the Kolka Glacier in the Russian Caucasus in 2002 (Evans et al., 2009), and the recent $68 \pm 2 \times 10^6$ m³ and $83 \pm 2 \times 10^6$ m³ collapses of two adjacent glaciers in the Aru range in the Tibetan Plateau (Tian et al., 2017; Gilbert et al., 2018; Kääb et al., 2018).

The massive, sudden detachments of both the Kolka and Aru glaciers caused the loss of human lives (Evans et al., 2009; Tian et al., 2017). These two extreme events have been critical in posing relevant questions on the origin and dynamics of massive glacier collapses of low-angle glaciers and their implication for glacier-related hazards over other mountain areas worldwide (Kääb et al., 2005). The recent Caucasus and Tibet events also showed that glacier instabilities of catastrophic nature with no historical precedents can happen under specific circumstances, and that previously known catastrophic glacier instabilities should be re-evaluated in the light of the new findings in order to investigate their relation to processes involved in the massive Caucasus and Tibet glacier collapses, or ice avalanching from steep glaciers, respectively (Kääb et al., 2018).

In this contribution we present the collapse of a cirque glacier in the Central Andes of Argentina in March 2007, which we informally named Leñas glacier. Owed to the isolated location of the glacier and the lack of human activity affected, the event had remained unnoticed until recently (Falaschi et al., 2018). Based on the analysis of aerial photos, high resolution satellite imagery and field observations, we follow the evolution of the Leñas glacier from the 1950´s through present day, describe the collapse event and later changes of the avalanche ice deposit, and discuss the possible triggering factors for the collapse. It should be noted that the remoteness of the study site, and the fact that the event remained unnoticed for a decade, limit the data base available to interpret the event. We consider it nonetheless important to report about this unusual glacier collapse in order to contribute to the discussion about glacier instabilities.

## 2 Study area



The Leñas glacier (34º 28' S – 70º 3' W; lower limit ca. 3450 m asl.; Fig. 1a) is located at the headwaters of the Atuel river, in the Argentinean province of Mendoza. The climate in this portion of the Andes of Argentina and Chile has been described as a Mediterranean regime. Snowfall maxima occur during the austral cold season (April-October), as the westerly flow drives frontal systems eastwards from the Pacific Ocean over the Andes. Glaciers in the Central Andes have retreated significantly since the second
half of the 20th century (Malmros et al., 2016). Specifically in the Atuel catchment, Falaschi et al. (2018) reported a moderate (though highly variable) glacier thinning rate of 0.24 ±0.31 m a$^{-1}$ overall for the 2000-2011 period.
Regarding glacier instability processes, there is a total of 16 glacier avalanches in the Tropical Andes of Peru and Colombia contained in the WGMS database (WGMS, 2017). In the classic work of Lliboutry
(1956) on the glaciers of the Southern Andes, a number of ice break-offs in icefalls in the Central Andes of Chile are reported, though none of them were out of the ordinary in order to have raised particular consideration. More recently, at least two glaciers in Central Chile have lost a significant portion of their mass in sudden collapses (Iribarren Anacona et al., 2014), namely, the $7.2 \times 10^6$ m$^3$ detachment of a debris-covered glacier just south of Cerro Aparejo (33°34'S–70°00'W) in March 1980 (Marangunic,
1997), and the 1994 ice avalanche in the southern flank of Volcán Tinguiririca (34°48'S–70°21'W), merely 50 km southwest of the Leñas glacier (Iribarren Anacona et al., 2014). A second, $10\text{-}14 \times 10^6$ m$^3$ ice-rock avalanche originating from Tinguiririca glacier occurred in January 2007 (Iribarren Anacona and Bodin, 2010; Schneider et al., 2011), only two months before the Leñas event.
The abundance of rock glaciers and perennial snow patches in the Leñas glacier surrounding (Fig. 1a)
indicates that permafrost is widespread in the area. Brenning (2005) indicated for the region that the minimum elevation of rock glacier fronts is indicative of negative levels of mean annual air temperature (MAAT) and of the altitudinal lower limit of discontinuous mountain permafrost, and set its extent at 3200 m in the nearby Cerro Moño range (34º 45' S; see also Brenning and Trombotto, 2006; Brenning and Azócar, 2010; Azócar and Brenning, 2010). This value is comparatively higher than the ~2800 m
elevation established for the whole Atuel catchment by IANIGLA (2015). The rough global permafrost zonation index map (Gruber, 2012) also indicates probable permafrost around Leñas glacier.
Lithology in the glacier surroundings is chiefly composed of pre and post-glacial volcanics (basalts, andesites and dacites) of Pliocene and Holocene age. Glacial, fluvial and mass removal processes have eroded and transported these rocks, which form the glacier forefields and outwash plains.


### 3 Satellite imagery and field observations

During the five decades prior to the 2007 collapse, Leñas glacier occupied a small glacier cirque, south
below the rockwall of the Morro del Atravieso peak (4590 m) and had a short debris-covered tongue in the flatter terrain underneath. Before the collapse, the elevation range of the glacier spanned between 4555-3441 m and the average slope was 24.6º. The analysis of available aerial photos shows that the glacier had an area of ~2.24 km$^2$ in 1955, and had shrunk to ~2.15 km$^2$ by 1970. There was no further area decrease until 2007. Concomitantly, the front retreated some 200 m between 1955-2007.
Sometime between 5 March (Landsat image showing an intact glacier) and 14 March 2007 (SPOT5 image showing the collapse), the lowermost part (3630-3441 m) of the glacier detached from the main glacier and produced an ice avalanche that ran down the valley for ~2 km, measured from the uppermost part of the scarp to the most distant point of the fragmented ice mass. Immediately after collapse, the ice avalanche had an area of 0.7 km$^2$. The orographic right (western) portion of the glacier subsided, but the
break-off was restrained by a lateral moraine. The altitude difference between the scarp head and the avalanche terminus of only 190 m results in a low angle of reach of only 5º (i.e. the so-called 'Fahrböschung'). The failed glacier area was about 250,000 m$^2$ as derived from SPOT5 imagery of 12 February 2007 and Quickbird imagery of 19 April 2007. The average glacier thickness at the scarp was roughly 35 m as estimated from scarp shadows and solar angles at the time of image acquisitions. Thus,
assuming linear decrease of the glacier thickness from the scarp to the former glacier front, an average glacier thickness of 18 m yields a rough estimate of $4.5 \pm 0.7 \, 10^6$ m$^3$ of ice detached in the avalanche with a conservative 15% error bar from uncertainty in detached area and thickness estimate.
For an independent estimate of the collapse volume we difference the February 2000 1-arcsecond C-band SRTM DEM and the ALOS PRISM World DEM AW3D (Fig. 2a). For the SRTM DEM we assume no
penetration of the radar pulse into the snowpack as February 2000 falls in austral summer with melting conditions likely. The assumption of no to little radar penetration is confirmed by the fact that the C-band and the X-band SRTMs show no significant vertical difference over the Leñas and other glaciers in the area. The AW3D DEM is stacked from individual DEMs from ALOS PRISM optical stereo triplets.





Exploration of the PRISM archive shows that the first suitable scene of the study site is from 18 April 2007 (i.e. after the collapse), and there are good scenes for every year over 2007–2011. In the averaged DEM product AW3D the elevations should thus roughly represent the average year 2009. The differences between the 2000 SRTM and the ~2009 AW3D DEMs (Fig. 2a) reveal a volume loss over the collapse detachment area of 4.3 ±0.3 $10^6$ m$^3$. This collapse volume agrees very well with the above estimate based on scarp shadows. Our latter DEM-based estimate does not take into account, though, glacier elevation

changes between 2000 and the collapse date in March 2007, and not changes of the detachment area between the 2007 collapse date and the date represented by the AW3D DEM. The volume error of ±0.3 Mio m$^3$ was calculated according to Wang and Kääb (2015) using an off-glacier standard deviation of elevation differences found to be 2.3 m and a conservative autocorrelation length of 400 m.

The large crevasses that would later delineate the collapse scarp were clearly visible 3 weeks before the

collapse (Fig. 3c) but strong crevassing at approximately the same location is also evident in the 1970 aerial photos (Fig. 3b). This indicates a potential break in slope of the bedrock at this location. Interestingly, the upper, steeper part of the glacier that had been mainly devoid of rock debris before the collapse, gradually became debris covered after the break off (Fig. 3e,f). It is however unclear if this development is related to the collapse (e.g., due to decrease in glacier slope), or coincidental (e.g., related

to overall glacier shrinkage in the area, or increased rock fall activity from the steep mountain flank above the glacier).

The ice avalanche deposit transformed from a mostly clean-ice surface directly after the collapse in 2007 (Fig. 3d) to a debris-covered one later on (Fig. 3e-f, 4a). Ice interspersed with rocks is featured at the avalanche terminus in the 19 April Quickbird image (Fig. 3d) and by 2011, the full ice debris (as most of

the upper portion of the glacier) had been sheltered by scree. Also, the detachment scarp and crevasses have disappeared, and large thermokarst ponds have formed within the avalanche deposit. The overall glacier slope decreased from 24.6º to 20.4º from before to after collapse. Currently, the avalanche terminus lies ~450 m up the valley with respect to the maximum avalanche extent in 2007.

The ice deposits of the Leñas collapse sit on a flat leveled plateau consisting of volcanic rocks, reworked

by glacial erosion, rockfall and maybe previous collapses. As stated above, the ice avalanche is meanwhile fully debris covered, though massive ice is visible on the walls of thermokarst ponds and ice cliffs. Within the avalanche deposit, which has now mostly a subdued and concave topography, at least two small outwash plains are forming, one at the abrupt slope change just above the uppermost reaches of the avalanche deposit, the other in front of the avalanche terminus (Fig. 4a).

Recent field observations of the detachment area done in March 2018 confirmed the absence of a hard bedrock underneath the glacier, as already suggested by the high-resolution satellite images. Also, the terrain under the former avalanche scarp is steep and not too rough (see Fig. 4a). Further down, debris in the ice avalanche deposit is composed of fragments of volcanic rock (<0.5 m in size) contained in a finer (pelitic to sandy) matrix, and very few large boulders (Fig. 3a,c). We assume this material to be further

evidence of the soft bed upon which the glacier rested before collapse. Between the outer limit of the ice avalanche and the LIA moraines (Fig. 1a), the terrain is made up of a chaotic arrangement of hummocks and thermokarst ponds that appear similar (though smoother) to the complex topography of the actual avalanche deposit. From the 2018 terrain inspection and the 1955 and 1970 aerial photographs, from befoe the collapse, it appears that the ice avalanche flowed over a seemingly bumpy, rough surface.


## 4 Meteorological and seismic data

We used the CHIRPS daily precipitation data (Funk et al., 2015), with a spatial resolution of 0.05° to

identify unusually high rainfall occurrences. During the period between 4-15 March 2007, no precipitation was recorded in the CHIRPS pixel where the Leñas collapse occurred and its surrounding pixels. These results were further verified with data from in-situ observations from the Laguna Atuel meteorological station. In addition, daily temperature reanalysis fields from ERA-Interim (Dee et al., 2011) were analyzed, considering the anomalies over the study area based on the 1981-2010 standard

period. Results show that temperature anomalies close to 3°C above normal were recorded during 11 and 12 March 2007.

Using data from the USGS earthquake catalogue (http://earthquake.usgs.gov) and applying the ground acceleration criteria discussed in Kääb et al. (2018) we find no earthquake between 4 and 15 March 2007 that could have triggered the Leñas collapse. The strongest earthquake found during the period of concern

and within a radius of 1000 km had a magnitude of 5.0 and distance of about 200 km from Leñas (depth 35 km; 11 March). The closest earthquakes (20-30 km) had magnitudes of 2.5 (4 March, 8.3 km depth) and 3.2 (11 March, 128 km depth).





## 5 Discussion


In terms of volume and glacier and runout slopes, the type of the 2007 Leñas glacier collapse ($4.5 \times 10^6$ m$^3$) seems to range somehow in between the massive collapses 2002 in the Caucasus and 2016 in Tibet, and ordinary ice avalanches. Compared to the Kolka ($130 \times 10^6$ m$^3$; Evans et al., 2009) and Aru ($68 \pm 2 \times 10^6$ m$^3$ and $83 \pm 2 \times 10^6$ m$^3$; Kääb et al., 2018) collapses, the Leñas event has a much smaller mass of ice


sheared off due to a smaller and shallower glacier. On the other hand, the 2007 Leñas event is also not typical for regular ice avalanches as the glacier is not very steep (the detached glacier part has a 15.6º mean slope) and the event volume is at the upper margin of more typical ice break-offs (Failletaz et al. 2015; Alean, 1985).

As potential factors for large glacier collapses, a number of causes have been investigated so far, namely


(i) high liquid water input into the glacier system from precipitation and melting, (ii) seismicity, (iii) changes in glacier geometry, and (iv) a shift in the thermal regime towards warmer conditions (Gilbert et al., 2018; Kääb et al., 2018). In the first place, our analyses of meteorological data showed no evidence of unusually strong increases in precipitation or temperature in the days immediately preceding the Leñas collapse that would directly destabilize the glacier. Neither do earthquake records reveal any strong


seismic activity that could have triggered the collapse. Instability may be favored as a glacier recedes from a flatter foot back into a steeper part of the bed, loosing thus the fontal stabilization in a type of self-debuttressing process, as also found for some more typical ice break-off situations (Faillettaz et al., 2015). From the very slight glacier area decrease in the Leñas case (2.24 to 2.15 km$^2$) before collapse, we cannot identify a significant change in glacier geometry that would have changed its stress regime, but this


finding could in parts be due to the limited availability of suitable DEMs. As for a change in thermal regime, from the rock glaciers in the area and the long preservation of the collapse deposits (see below) we conclude a potentially cold ground temperature regime for parts of the glacier. The thin glacier front could have been frozen to the bed, and a change in this polythermal regime may have caused changes in stability.


An important finding from field work is the abundance of fine sediments in and on the collapse deposits (Fig. 3). We suggest that the soft glacier bed material could have played an important role in the collapse enhancing avalanche mobility, as already noted for the Kolka and Aru collapses (Gilbert et al., 2018). The rather short run-out distance in the Leñas collapse, however, may imply a rougher or more permeable glacier bed where no basal water layer could develop that would have reduced the friction of the glacier


compared to the Aru avalanches (Kääb et al., 2018; Gilbert et al., 2018). In addition, the short run-out distance could simply be due to the smaller collapse volume and thus smaller potential energy involved, and to flatter and more hummocky terrain.

We hypothesize a mixed origin for the debris layer observed on the ice avalanche deposit. On the glacier head, frost action and permafrost thaw are probably responsible for the production of fine grain deposits


originating from rock fall off the steep and ice-free surrounding rock walls (Fig. 3, panels e and f). The compact pieces of ice with a small amount of debris on top of them (Fig. 4b) may be intact parts of the former debris-covered glacier front that detached as a whole (or in few large fragments) and formed the front of the collapsed ice mass (cf. Fig. 1b, 2d). The loss of the glacier front likely debuttressed higher (and not debris-covered) glacier parts that came down after the front, either in direct sequence or even


with some delay, in the latter case suggesting the possibility for different phases of the collapse with different properties. The former glacier front might have also ploughed through the forefield and in parts have taken up debris there together with the original debris cover on the glacier front. As the ice avalanche ablated during subsequent years, debris concentrated further on top of the deposit (Rowan et al., 2015).


The notably irregular terrain over the forefield (entire forefield before 2007, or outside avalanche deposits after 2007, respectively) may indicate that a large glacier collapse has not happened in 2007 for the first time. This speculation relies, though, on a link between ice avalanche deposits from Leñas glacier and the forefield terrain characteristics, which we have only some visual indications for at this point.

In support of the permafrost indications given in section 2, the ice-cored moraines and rounded features in


the glacier forefield (Fig. 4d), point to ground ice deformation under permafrost conditions. Because the flat plateaux is distant from the rockwalls, where the terrain is steep and protected from incoming solar radiation, we speculate that most of it is permafrost-free and the ground ice beneath the upper soil layer could be a remnant from a previous, much larger ice avalanche. A further sign of permafrost conditions below the avalanche deposit, favoring the preservation of the avalanche ice, is the rather good


preservation of much collapse ice in the ice-debris deposit, even 11 years after collapse. In comparison, the ice of the Aru glacier collapses will have melted away to a large extent 2 years after collapse (Kääb et al., 2018), whereas the heavily debris-covered and up to more than 100 m thick deposits of the Kolka glacier collapse lasted many years despite their low elevation, far below the local permafrost limit.





As the Tinguiririca event (see Introduction) happened only a few weeks before the Leñas glacier collapse, and questions about similar causes or triggers might arise, we want to also shortly summarize this event. However, though spatially and temporally close, the Leñas and Tinguiririca events are probably different in nature. In the first place, the Tinguiririca event involved a much larger volume (10-14 × $10^6$ m$^3$ vs. 4.5 × $10^6$ m$^3$ -Schneider et al., 2011).. From DEM differencing (Fig. 2b), we re-estimated a volume of 15.6 × $10^6$ m$^3$, using the same method as for the Leñas event. Secondly, the source slope is a bit higher (~20° vs.
15.6°), and, most important, instead of a fine, soft bed, the glacier was lying on a hard bed (Fig. 2c; see also Iribarren Anacona et al., 2015). Also, neither the rigid bedrock (as opposed to a fine-grained sediment bed) nor the very low water content involved did impede great mobility (8.2 km, more than four times the 2 km Leñas run-out distance over a ~10° angle of reach) in the Tinguiririca ice avalanche (Iribarren Anacona and Bodin, 2010). It was rather the rock bedding planes being nearly-parallel to
bedrock surface and smooth surface that provided ow friction conditions at the base of the avalanche (Schneider et al., 2011). The Tinguiririca ice-rock deposit was lying in a presumably non-permafrost area below 3000 m, and has been washed away by fluvial erosion ever since. This means it is not possible anymore to retrieve and compare the detailed geomorphic and petrology characteristics between both events.


## 5 Conclusions

In the region of the Central Andes studied here, gravity-driven failures of steep glaciers have been previously observed. The volume of the Leñas collapse of ~4.5 $10^6$ m$^3$ and the detachment slope of 15.6°, however, deviate from the more typical ice avalanches from steep glaciers and place the event closer to low-angle glacier collapses. Due to the large time difference between the Leñas glacier collapse in 2007 and its discovery, and the remoteness of the site, only limited data are available to analyze the case. As a main finding, the event does not rule out the importance of special soft bed characteristics as a common
factor in the (rare) collapse of low-angle glaciers (Kääb et al., 2018; Gilbert et al., 2018). We are not able to identify a clear potential trigger of the Leñas event, as neither the meteorological or seismic data reveal unusual conditions or events that could have triggered the Leñas collapse, nor a significant change in glacier geometry could be identified. Even though we have no strong evidence, we cannot rule out that similar collapses from Leñas glacier happened before. Despite the knowledge deficiencies related, for
instance, to the hydrological, hydraulic, or ground-thermal conditions under which the Leñas glacier collapse took place, the information presented here adds to the spectrum of environmental and glaciological circumstances under which glacier collapses can take place, including related implications for mountain hazard management.


**Author contribution**
DF led and designed the study, conducted the field work, analyzed data, and wrote the paper. AK and FP helped in designing the study, analyzed data, and wrote the paper. TT processed and orthorectified the ALOS PRISM imagery. JAR prepared and analyzed the meteorological station and reanalysis data. LL
helped in designing the study.

**Competing interests**
The authors declare that they have no conflict of interest.

**Data availability**
Landsat data and the SRTM C-band DEM are available from http://earthexplorer.usgs.gov. The SRTM X-band DEM is available from http://eoweb.dlr.de. The ALOS AW3D is available from http://www.eorc.jaxa.jp/ALOS/en/aw3d30/. Earthquake data are available from http://earthquake.usgs.gov. Data from DigitalGlobe (Quickbird) and Airbus (SPOT) are commercial.


**Acknowledgements**
The present study was carried out in the framework of the SeCTyP project *Investigación, a partir de las Aplicaciones Geomáticas de los cambios recientes en los ambientes glaciarios relacionados con la variabilidad climática en las cuencas superior del Río Mendoza y del Río Atuel* from the Universidad
Nacional de Cuyo, Argentina.
The authors would like to thank Dario Trombotto (IANIGLA) for his comments on permafrost and Mariana Correas Gonzalez, Andrés Lo Vecchio (IANIGLA) and the *baqueano* Saul Araya for assistance during field work. Andreas Kääb acknowledges support by the European Research Council under the



European Union's Seventh Framework Programme (FP/2007–2013)/ERC grant agreement no. 320816, and the ESA projects Glaciers_cci (4000109873/14/I-NB) and DUE GlobPermafrost (4000116196/15/IN-B).

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




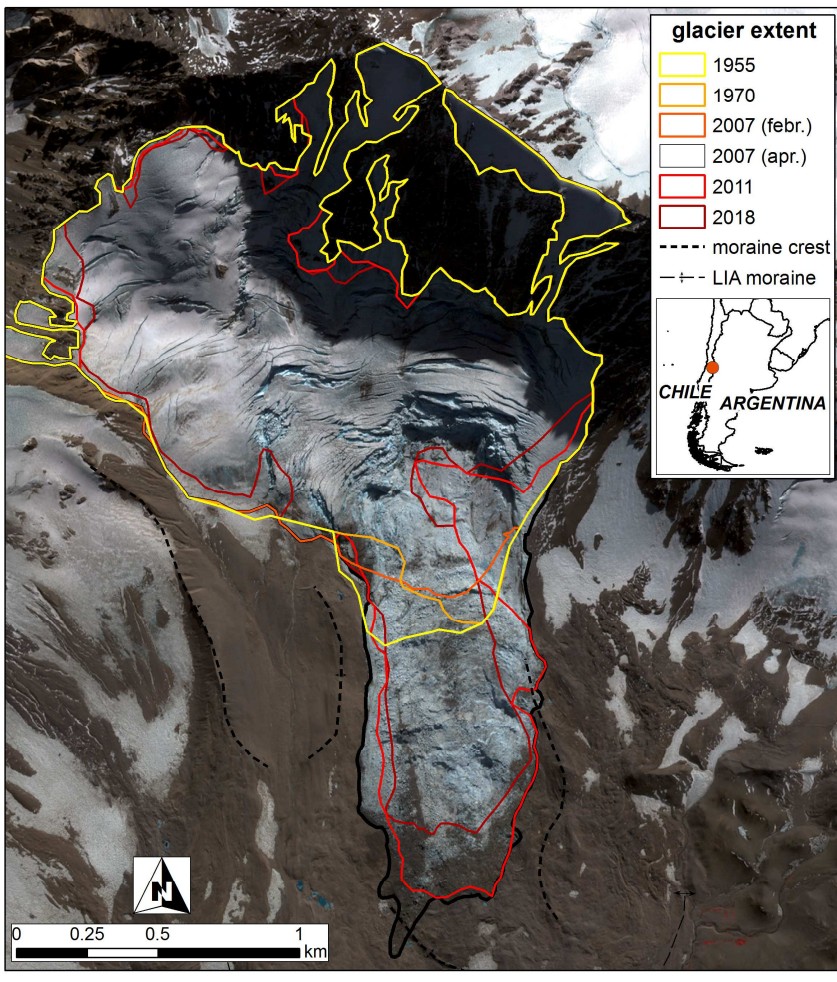

**Figure 1: Quickbird image of the Leñas avalanche from April 2007, after collapse, and glacier frontal positions for the 1955-2018 period. Inset: Location of the Leñas glacier in the study area.**




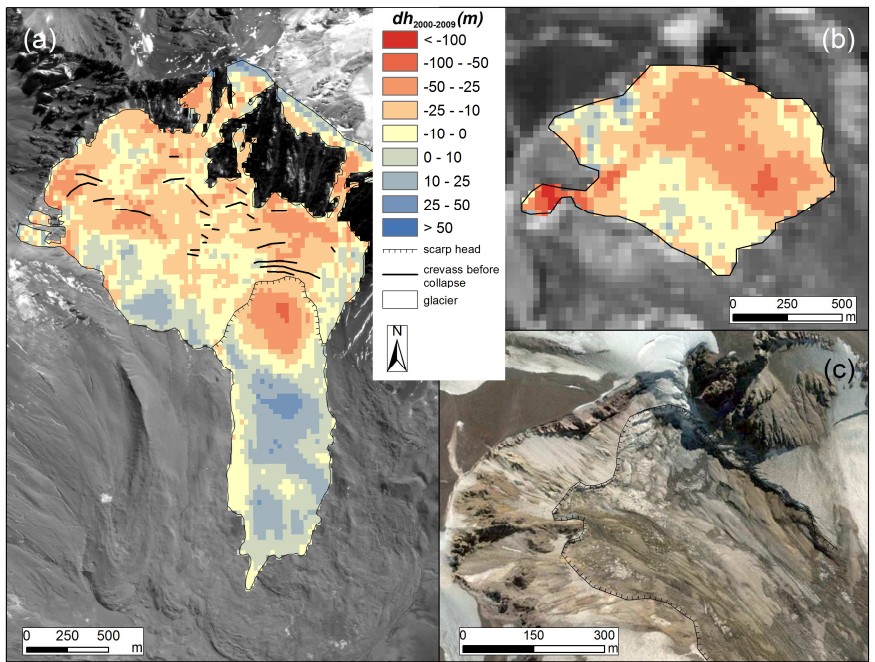

**Figure 2: 2000-2009 elevation differences in the Leñas (a) and Tinguiririca (b) collapses. Backround images in (a) and (b) are ALOS PRISM and Landsat. (c) shows the scarp and detachment area of the Tinguiririca ice-rock avalanche (Google Earth).**






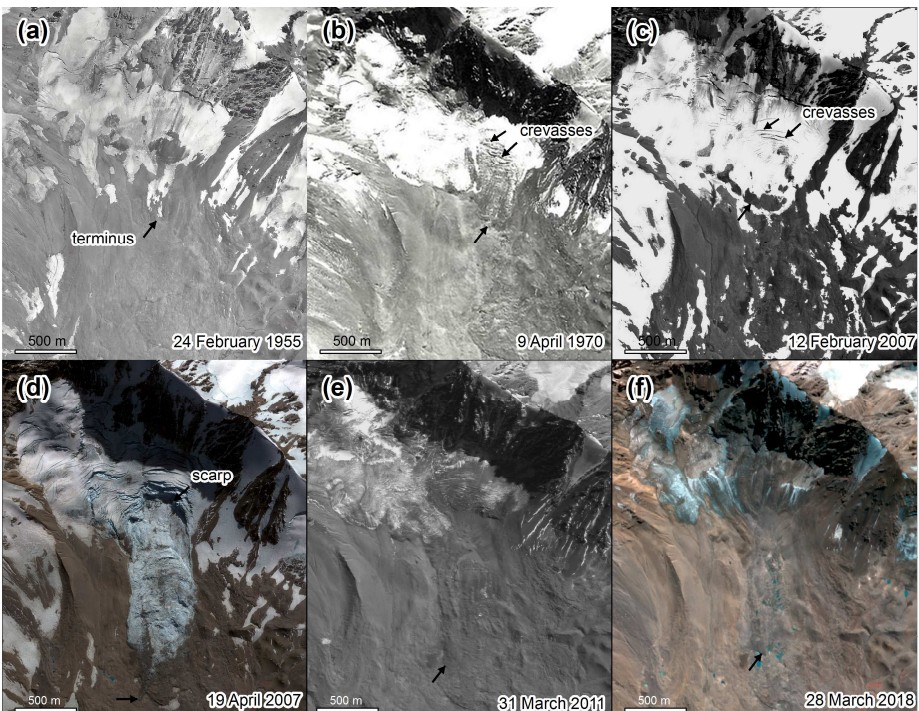

**Figure 3: Evolution of glacier extent through time, the avalanche (d) increasing debris cover (e, f). The terminus position of the former glacier (a –aerial photo) and the subsequent avalanche fronts are marked with a black arrow on all panels. The large crevasses visible in the 1970 aerial photo (b) and the SPOT 5 image from February 2007 (c) demark the location of the scarp head in the Quickbird scene of April 2007 (d). (d), (e - ALOS PRISM) and (f -Planet) depict the growth of debris-covered portions on the glacier and the transformation of the collapse deposits.**


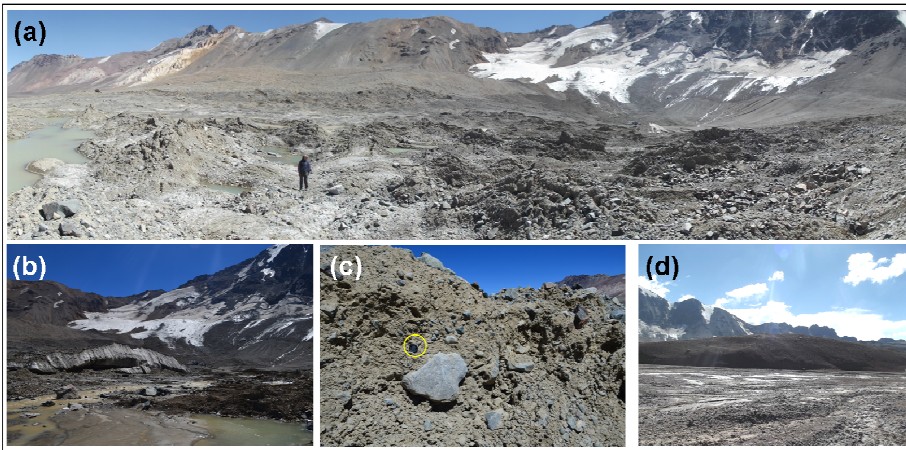

**Figure 4: (a) Panoramic view of the Leñas glacier and avalanche deposit in March 2018, with the glacier head on the far upper right. (b) Former glacier fragment at the base of the detachment area arrow in Fig. 3). (c) Detail of the debris cover on the avalanche deposit, showing the rock fragments and matrix (see the black camera objective cover inside the yellow circle for scale). (d) Presumably ice-cored LIA moraines.**

