# Peer review of "Brief communication: Collapse of 4 Mm3 ice from a cirque glacier in the Central Andes of Argentina"

_The Cryosphere, 2018_

## Referee Comment (RC1) · Anonymous Referee #1 · 31 Oct 2018

General comments :

This study deals with the collapse of an unnamed glacier in the central Andes of Argentina in March 2007 (between 5 March and 14 March 2007). This glacier, named by the authors, ' Lenas glacier ', is located in the very remote area. The collapse and the avalanche have not been observed directly and it seems to remain unnoticed during several years. Very few data are available about this collapse. Most of data come from satellite images. This study aims at reconstructing the conditions of the collapse (Volume, slope, meteorological conditions, sismic events,….) in order to understand the possible causes of this breaking off. The authors claim that this event, very rare, can be compared with the very large collapses of Kolka in 2002 and Aru glaciers in 2016, given that the volume size is huge and the slope of the glacier is low.

[Figure]

Unfortunately, the analysis and the conclusions remain qualitative and speculative due to the lack of data. This study contains vagueness, large assumptions, and lack of rigour for the following reasons :

1) First, the uncertainty relative to the collapse volume can be questioned. The volume changes have been assessed from satellite images (Spot 5 12 February 2007, Landsat 5 March 2007, Spot 5 14 March 2007, Quickbird 19 April 2007). The failed glacier area and thickness have been estimated from images 12 February 2007 and 19 April 2007. The accuracy of each DEM is not mentioned. The authors wrote that ' the average thickness at the scarp was roughly 35 m as estimated from scarp shadows and solar angles at the time of acquisitions . Thus assuming a linear decrease of the glacier thickness from the scarp to the former glacier front . . . .. a rough estimate of 4.5 106 m3 ' (l. 108-111) . No detailed information is given about the method and the uncertainty of thickness. In the following lines (l.112), the authors mentioned a ' conservative 15% error from uncertainty in detached area and thickness estimate' without providing any details about this uncertainty. Another ' independent ' estimate has been done from the difference between SRTM DEM (February 2000) and ALOS PRISM DEM AW3D obtained between 2007 and 2011. Again, the uncertainty of these DEM is not given. The uncertainty related to SRTM penetration is not mentioned. The authors mentioned only that the assumption of no radar penetration is confirmed by the comparison between C Band and X Band SRTM which show no significant difference. The authors did not provide any detail or reference. In addition, the authors used the ' average DEMS' of ALOS PRISM DEM (2007-2011) with an ' average year' of 2009. They assume that there is no change between March 2007 and 2009 (l. 125-126). It is very confusing. The uncertainties relative to this assumption are not explained. Nothing is said about the elevation differences between these 2007-2011 DEMs. The uncertainty of 2.3 m (l. 128) on elevation difference seems to be very optimistic. Moreover, from Figure 2a, one can see surprising elevation changes of 25-50 m in several areas of the upper part of the glacier between 2000 and 2009 far from the detached zone. These values are similar to the elevation changes of the collapse

area. However, no explanation is given about that. Due to the lack of information, it seems very difficult to assess the uncertainty on the volume of the collapse.

2) The discussion about the ' mean slope' is confusing. The authors make a difference between hanging glaciers with steep slopes (>30°) and glaciers with low slope (lines 26-27). They wrote that ' the detachment of large portions of low-angle glaciers is much less frequent ' (l. 34-35). The manuscript is confusing because the authors mentioned both the low angle of reach (5°) (lines 22 and 106), the average slope of glacier (24.6°) (line 97) and the slope of the detachment part to discuss the stability/instability of the glacier. These slopes are mentioned in different sections of the manuscript which creates confusion. The low angle reach is not relevant to study the stability of hanging part of the glacier. More specifically, the ' suprisingly low angle of reach (5°) ' (Abstract, l. 22) seems to be irrelevant as an indicator of stability of glacier. The slope of ' detached glacier', which seems to be the relevant value to assess the stability, is mentioned in Discussion only in line 186 (15.6°). We do not have any information about the method used to calculate this slope. Is the surface slope before the collapse ? calculated on which distance ? is the surface slope after the collapse ? Which images have been used to obtain this value ? What is the accuracy of this calculated value ? The analysis of slope change reveals also a lack of rigour. In line 142, the authors wrote : Âń the glacier slope decreased from 24 .6 to 20.4° from before to after collapse Âż. The distance on which this slope is calculated is not specified. One can assume that the slope change is mainly due to the length changes of the glacier and the size of the avalanche. In this way, is the slope change a relevant information?

3) As mentionned by the authors in Conclusions, due to the limited data, this study is not able to identify the causes of the Lenas event. Many assertions are highly speculative. For example ' the thin glacier front could have been frozen to the bed and a change in this polythermal regime may have caused changes in stability' (l. 202-204) or ' we suggest that the soft glacier bed material could have played an important role in the collapse. . ..' (l. 206-207), or ' we hypothesize a mixed origin for the debris layer

observed on the ice avalanche deposit. . .' or ' . . .may indicate that a large glacier collapse has not happened in 2007 for the first time. This speculation relies on. . . . . . '. The Discussion is a list of assumptions and questions and does not shed light on the causes of this collapse.

In summary, the authors claim that the Lenas collapse deviates from typical ice avalanches from steep glacier and can be compared to the rare low-angle glaciers collapses similar to Aru glaciers and Kolka glaciers avalanches. Given the lack of information given in this study, the uncertainty on collapse volume can be questioned. In addition, the data provided by this study are poor and do not allow to identify the possible causes of the collapse. This study points out the low detachment slope (15.6°) although the determination of the slope has not been explained and the uncertainty on this slope is unknown. 'No significant change in glacier geometry could be identified due to the lack of data' as mentioned in Conclusions (l. 267-268). The authors suggest that soft bed characteristics play a crucial role on the collapse trigger. I do not think that this study provides sufficient quantitative information for understanding complex processes in glacier instabilities and collapses. I do not think this study shed new light on the triggers and factors responsible for this event. Given the paucity of data, I am not sure that this event can be compared to Aru glaciers and Kolka glaciers avalanches as claimed by the authors.

---

## Referee Comment (RC2) · Walter (Referee) · 12 Dec 2018

This submissione documents the catastrophic collapse of a glacier tongue in the Central Andes. The volume can be considered small compared to the well known 2016 twin collapse of Tibetan glaciers (around an order of magnitude smaller) but large compared to ice avalanches typically encountered in Alpine terrain. The described event dates back to 2007 and occurred in a remote area, consequently little evidence is available and the analysis evolves around satellite-derived digital elevation models and images.

The study cannot provide much physical insights into the processes leading up to the collapse, but this should not be expected given the relatively sparse catalog of observations. On the other hand, the 2016 glacier collapses in Tibet vividly illustrated

that catastrophic runaway surges of low-angle glacier tongues can occur and may be related to climate, a matter that previously had been largely overlooked by the glaciological community. So I agree with the authors that this short note contribution is of interest to the glaciological community and well suited the Cryosphere journal.

To my mind the manuscript requires a considerable amount of modifications, however. In particular, several figures are poorly presented, annotated and referred to in the text, which obscures some important information on the collapse event. Some of my points of criticism may be misunderstandings on my side but I nevertheless urge the authors to consider and clarify them and make the necessary adjustments to convey their message in this short note. Below I detail these points and provide further minor questions and comments. Fabian Walter.

MAJOR COMMENTS FIGURES I have to admit I was puzzled when looking at the details of Figures 1-3. It may sound picky, but I was confused because I did not understand the authors' conception of the glacier outlines. Usually I think of avalanche debris as not being part of the glacier, i.e. a calving event (dry or wet calving) causes the glacier to retreat. If I understand the outlines in Figure 1 correctly, then the authors consider the avalanche debris as lying within the glacier extent. Whatever the case is, this should be clarified and my personal suggestion is to define the glacier outlines by the ice that has NOT detached from the glacier in the form of avalanches. Moreover, in Figure 1, there are several grades of red lines, which are difficult to distinguish on the image. It would be better to use different colors or line symbols. The figure would also benefit greatly from two panels, one showing the glacier before and one showing the glacier after the collapse. Within the figure I would also label the avalanche debris as well as the LIA moraine, which is discussed in the text. Once this is clarified, it will be easier to understand Figure 2. Here I was wondering for a long time why the Leña Glacier tongue had thickened. Is this a result of surging behavior? Then I noticed that what I thought was the tongue was actually the avalanche debris. It would help to see the extent of the glacier tongue before the break-off in addition to the shown

scarp head (note also that some of the text in this figure is likely too small). Similarly, in the different panels of Figure 3 I suggest drawing the glacier outlines. Finally, the photographs in Figure 3 are not self explanatory but do seem to contain important information. I suggest annotating the photographs extensively (e.g. glacier terminus, LIA moraine, avalanche debris, outwash planes, etc.).

TINGUIRIRICA GLACIER Compared to Leñas glacier, Tinguiririca Glacier receives less attention in the text. It is only illustrated in two panels of Figure 2. The reader needs a map view equivalent to Figure 1 to get a feel for the glacier extent and geometry (for both glaciers it would be helpful to see a few contour lines which helps identifying steep parts and planes) and more explanations, otherwise it seems that Tinguiririca Glacier was half-heartedly added to the study.

SURGE HISTORY The topic of glacier surging receives little attention in the manuscript. Do the satellite DEM's provide some hints for surge behavior? In any case, it would be good to write 1-2 sentences on this subject to put the collapse into context of the Aru Co and Kolka events. This could be built into the second paragraph of the Discussion section. Currently, there is some mentioning of a thermal regime change, but no specific evidence or context is provided.

MINOR COMMENTS Figure references: At several parts of the manuscript it is not clear what the autor's assertions are based on. For example, in the first paragraph of Section 3 no references to figures are made, but if I understand correctly the described observations are based on information shown in the figures.

Line 68: It would help to give a rough estimate of avalanche volumes for the 16 events of the WGMS.

Line 134: "(e.g. due to decrease in glacier slope)" is unclear. Line 140-141: Do detachment scarp and crevasses really disappear or were they simply covered by debris?

Near Line 150: How was the absence of bedrock beneath the glacier confirmed? Using

boreholes? Could exposed bedrock be concealed by deposited sediments? Please mark/annotate figures accordingly.

Line 152: Reference to Figure 4a is unclear. Please mark/annotate figure accordingly.

Line 156-157: The smoother hammocks and thermocarst should be highlighted in the respective figures.

Section 4: It would help to show parts of the meteoroligical analysis in a plot. Also, some specifics on the acceleration criteria would be of interest.

Line 183 (and elsehwere): It would help the non-expert to specify what is mean by "ordinary" ice avalanches.

Discussion: It may be worth considering the possibility that the event happened as a series of small break-offs rather than a single rupture. Such cases are known to exist and it is not clear which conditions favor one scenario over the other. https://www.geopraevent.ch/project/weissmies-glacier-velocities/?lang=en

Line 229 and following paragraph: When discussing the permafrost conditions it seems that the authors present arguments for and against permafrost. It was not clear to me what the actual conditions are believed to be. Also, it is not clear what the implication of the last sentence is (reference to Kolka).

Line 244: Specify "same method".

Line 262: "time difference" → time lag.

---

## Referee Comment (RC3) · Anonymous Referee #3 · 20 Dec 2018

**1. Summary**

Falaschi and coauthors describe an apparent sudden collapse of a small cirque glacier in the central Andes, Argentina. They describe this event largely through the use of remote sensing and limited field surveys. Although largely documentary in nature, this brief report provides additional evidence for sudden failure of alpine glaciers – these events present an unusual type of mountain hazard that may occur with greater frequently in the years ahead. Overall, I found the evidence for the failure convincing, and the manuscript's organization made their arguments mostly easy to follow. The paper will require moderate revisions to bring it up to the level required for publication in The Cryosphere, however. Below, I outline major points I have with the manuscript. I also include a marked up copy of the manuscript to help the coauthors revise their

paper.

**2. Major Points**

A) Remove or substantially trim speculative material - This manuscript represents a short note (Brief Communications) in The Cryosphere, and simply providing sound documentation for the event is sufficient for publication (largely because of the hazard implications associated with these events). In many places, however, the authors tend to go a bit too far in the interpretation of their data that leads to too much speculation (mostly about whether event was enhanced by fine-grained bed or whether events happened before). The authors point out, for example, that they have limited field observations, but then they go on to make claims that really require additional field observations or require modeling data that they currently to do not have in hand (or report in the paper). I think a proper documentation of the event is enough. Simply state factors that caused the event to happen are currently uncertain. The logic of several statements are flawed at least in the way they are written. (e.g. towards the end of the paper '...no strong evidence, but can't rule out past events...')

B) Stick to the event described in the methods/study area – In the discussion section the authors (line 240+) describe methods used to study another glacier collapse, but this site really wasn't described except primarily in the introduction of the paper. A reader can't really evaluate the evidence for that event as it now stands since it's only briefly described in the introduction of the paper. I would recommend the paper be revised to either describe both events or simply to refer to the other one in passing (in the present version of the paper the authors start to tell us about their DEM differencing and uncertainties in the discussion section of the paper).

C) Check co-registration/uncertainty analysis – The authors co-registered their DEMs prior to differencing, but I would request that they check the quality of that co-registration (see artifacts near top of cirque in Figure 2a). It may simply be as good as they can get, but some explanation for this offset over steep terrain would be useful.

D) English needs to be improved –The paper includes many statements that are unclear or overly vague. I would suggest that the second and third authors spend some time with the text to improve the English. There are also many topographical errors in the manuscript. These errors really should have been cleaned up prior to submission.

3. Figures and Tables

Figure 1. Latitude/longitude (even two) needed so one can locate this glacier. Also, please state which bands (spectral range) were used for the color composite.

Figures 2 and 3. I found the order of these figures to be reversed. I would first report on Figure 2 as this shows when the event happened. Figure 3 really is a derivative product of stereo imagery, so DEM make more sense to show after you introduce Figure 2.

4. References

I did not check the references for consistency, but this should be done on the revised paper.

5. Title – It's always awkward to start a title (or sentence) with a number. Why not just, 'Sudden collapse of

---

## Author Comment (AC1) · 15 Feb 2019

Summary
Falaschi and coauthors describe an apparent sudden collapse of a small cirque glacier in the central Andes, Argentina. They describe this event largely through the use of remote sensing and limited field surveys. Although largely documentary in nature, this brief report provides additional evidence for sudden failure of alpine glaciers – these events present an unusual type of mountain hazard that may occur with greater frequently in the years ahead. Overall, I found the evidence for the failure convincing, and the manuscript's organization made their arguments mostly easy to follow. The paper will require moderate revisions to bring it up to the level required for publication in The Cryosphere, however. Below, I outline major points I have with the manuscript. I also include a marked up copy of the manuscript to help the coauthors revise their paper.
*We thank the referee for the constructive comments. We have modified and corrected the manuscript following the comments and suggestions.*

2. Major Points A) Remove or substantially trim speculative material - This manuscript represents a short note (Brief Communications) in The Cryosphere, and simply providing sound documentation for the event is sufficient for publication (largely because of the hazard implications associated with these events). In many places, however, the authors tend to go a bit too far in the interpretation of their data that leads to too much speculation (mostly about whether event was enhanced by fine-grained bed or whether events happened before). The authors point out, for example, that they have limited field observations, but then they go on to make claims that really require additional field observations or require modeling data that they currently to do not have in hand (or report in the paper). I think a proper documentation of the event is enough. Simply state factors that caused the event to happen are currently uncertain. The logic of several statements are flawed at least in the way they are written. (e.g. towards the end of the paper '. . .no strong evidence, but can't rule out past events. . .')

*We agree that the nature of some statements is speculative when considering the available evidence and have thus shortened or removed them (e.g. the Tinguiririca avalanche). However, we think that it is allowed in the discussion section - if properly marked - to add some interpretation that is more speculative in nature. This should not only make clear that we have indeed recognized the geomorphological evidence of a probably larger previous collapse (to give one example), but should also identify open issues and point to interesting further studies. By presenting them here, there is a possibility to link potential future research proposals to such open issues and investigate them further. A pure observational report without reflecting about the lessons learned and open issues (i.e. the more speculative elements) would likely not warrant publication in The Cryosphere.*

*Most importantly, we have considerably shorten/fully removed most of the speculative ideas that were originally included in the discussion chapter (namely the influence of the fine grained material of the glacier bed, the presence/absence of permafrost conditions below the avalanche deposit and the possibility of previous collapses), trying to stick to the main event (Leñas collapse). As far as the latter is concerned, we have also limited the discussion*

*regarding the Tinguiririca avalanche due to the even lesser amount of available remote sensing data, removing the corresponding figure as well.*

B) Stick to the event described in the methods/study area – In the discussion section the authors (line 240+) describe methods used to study another glacier collapse, but this site really wasn't described except primarily in the introduction of the paper. A reader can't really evaluate the evidence for that event as it now stands since it's only briefly described in the introduction of the paper. I would recommend the paper be revised to either describe both events or simply to refer to the other one in passing (in the present version of the paper the authors start to tell us about their DEM differencing and uncertainties in the discussion section of the paper).

*We agree with the referee that the Tinguiririca avalanche was half-heartedly discussed in the study, and that there is even less remote sensing data available (high resolution satellite imagery, DEMs) for this event. Hence, we have removed the Tinguiririca event more or less completely (incl. the figures). It is now only briefly presented in terms of avalanche volume and runout distance.*

C) Check co-registration/uncertainty analysis – The authors co-registered their DEMs prior to differencing, but I would request that they check the quality of that coregistration (see artifacts near top of cirque in Figure 2a). It may simply be as good as they can get, but some explanation for this offset over steep terrain would be useful.

*In steep terrain it is indeed possible that DEMs have artifacts that are much larger than the real changes. However, they have no impact on the general pattern of the elevation trends observed here. Moreover, the causes and problems of such artefacts have been discussed widely in the literature and we have thus only added a short explanation and some further references.*

D) English needs to be improved –The paper includes many statements that are unclear or overly vague. I would suggest that the second and third authors spend some time with the text to improve the English. There are also many topographical errors in the manuscript. These errors really should have been cleaned up prior to submission.

*We apologize for these errors (assuming you mean typographical?) and will give the ms a proper English check before re-submitting it. Also, all TC papers undergo language editing by the publisher after acceptance.*

3. Figures and Tables Figure 1. Latitude/longitude (even two) needed so one can locate this glacier. Also, please state which bands (spectral range) were used for the color composite.
Figures 2 and 3. I found the order of these figures to be reversed. I would first report on Figure 2 as this shows when the event happened. Figure 3 really is a derivative product of stereo imagery, so DEM make more sense to show after you introduce Figure 2.

*Figures 1 and 2 have been reorganized according to another reviewer's suggestion and are now including coordinates. Figures 1 and 2, show the Leñas glacier (February 12, 2007 and April 19 2007) before and after collapse and the DEM differencing-elevation change map. We thought that looking at the glaciers before/after collapse together with the elevation difference map would help the reader to better interpret the event. We have included a latitude/longitude grid in figures 1, and have added to the text the RGB composites used.*

4. References I did not check the references for consistency, but this should be done on the revised paper.

*Thank you for noting. We have double-checked and added some more reference to the revised manuscript.*

5. Title – It's always awkward to start a title (or sentence) with a number. Why not just, 'Sudden collapse of

We agree and have now written: "Collapse of 4 Mm3 ice from a cirque glacier in the Central Andes of Argentina".

---

## Author Comment (AC2) · 15 Feb 2019

This submission documents the catastrophic collapse of a glacier tongue in the Central Andes. The volume can be considered small compared to the well known 2016 twin collapse of Tibetan glaciers (around an order of magnitude smaller) but large compared to ice avalanches typically encountered in Alpine terrain. The described event dates back to 2007 and occurred in a remote area, consequently little evidence is available and the analysis evolves around satellite-derived digital elevation models and images.

The study cannot provide much physical insights into the processes leading up to the collapse, but this should not be expected given the relatively sparse catalog of observations. On the other hand, the 2016 glacier collapses in Tibet vividly illustrated that catastrophic runaway surges of low-angle glacier tongues can occur and may be related to climate, a matter that previously had been largely overlooked by the glaciological community. So I agree with the authors that this short note contribution is of interest to the glaciological community and well suited the Cryosphere journal.

To my mind the manuscript requires a considerable amount of modifications, however. In particular, several figures are poorly presented, annotated and referred to in the text, which obscures some important information on the collapse event. Some of my points of criticism may be misunderstandings on my side but I nevertheless urge the authors to consider and clarify them and make the necessary adjustments to convey their message in this short note. Below I detail these points and provide further minor questions and comments. Fabian Walter.

*We thank Fabien Walter for the critical and constructive comments to improve the paper and have modified and corrected the manuscript accordingly. Also the misunderstandings served to clarify concepts in the manuscript (e.g. the glacier and ice avalanche deposit separation) and were useful to prepare better figures. The figures have also been re-arranged and further improved. Because parts of the manuscript have been rewritten, some of the comments are now obsolete (e.g. specific comment in line 229).*

MAJOR COMMENTS FIGURES I have to admit I was puzzled when looking at the details of Figures 1-3. It may sound picky, but I was confused because I did not understand the authors' conception of the glacier outlines. Usually I think of avalanche debris as not being part of the glacier, i.e. a calving event (dry or wet calving) causes the glacier to retreat. If I understand the outlines in Figure 1 correctly, then the authors consider the avalanche debris as lying within the glacier extent. Whatever the case is, this should be clarified and my personal suggestion is to define the glacier outlines by the ice that has NOT detached from the glacier in the form of avalanches.

*This is actually fully correct but would be unnoticed in a glacier inventory when not checking back the time series of high resolution images. In a 10 to 30 m satellite image (Landsat, ASTER, Sentinel 2) the collapsed (clean ice) part would very likely be mapped as a part of the glacier and the entire feature classified as a valley glacier. As regenerated glacier parts are typically included in glacier inventories and the discussion if this makes sense or not can be endless, the inclusion of the collapsed part is maybe not that wrong. However, as we are presenting a collapse event here, the collapsed part has been marked separately in the revised figures.*

Moreover, in Figure 1, there are several grades of red lines, which are difficult to distinguish on the image. It would be better to use different colors or line symbols. The figure would also benefit greatly from two panels, one showing the glacier before and one showing the glacier after the collapse. Within the figure I would also label the avalanche debris as well as the LIA moraine, which is discussed in the text. Once this is clarified, it will be easier to understand Figure 2.

Here I was wondering for a long time why the Leñas Glacier tongue had thickened. Is this a result of surging behavior? Then I noticed that what I thought was the tongue was actually the avalanche debris. It would help to see the extent of the glacier tongue before the break-off in addition to the shown scarp head (note also that some of the text in this figure is likely too small). Similarly, in the different panels of Figure 3 I suggest drawing the glacier outlines. Finally, the photographs in Figure 3 are not self explanatory but do seem to contain important information. I suggest annotating the photographs extensively (e.g. glacier terminus, LIA moraine, avalanche debris, outwash planes, etc.).

*We have prepared a new set of figures paying attention to the referee's comments and suggestions. Figure 1 contains three panels, showing the Leñas glacier before and after collapse, and the DEM differencing-elevation change map. 100m contour lines have been included. We have followed the referee's suggestion and clearly separated the glacier extent and the ice avalanche deposit. This is valid not only for the figure but for the area calculations in the main manuscript as well (the avalanched area is only 0.63 $km^2$ instead of 0.7 $km^2$ now). The glacier outlines for the years 1955, 1970, 2011 and 2018, as requested, have been removed from figure 1 to avoid confusion and transferred to figure 2, which effectively follows the glacier and avalanche deposit's evolution through time. Also, the LIA moraines can now to be seen more clearly in the figure.*

*We have eliminated the figure showing Tinguiririca glacier, as we have taken the decision to strictly stick to the event described in the introduction (see response to the Tinguiririca glacier comment below).*

*Figure 2 includes the 1955, 1970, 2011 and 2018 glacier and avalanche debris outlines, and has been annotated with (LIA) moraines, rock glaciers, glacier forefield, crevasses, avalanche scarp. We note that some of the avalanche features (thermoklarst ponds, hummocks, etc.) are too small to be annotated in this figure and have been marked in figure 3a instead.*

TINGUIRIRICA GLACIER Compared to Leñas glacier, Tinguiririca Glacier receives less attention in the text. It is only illustrated in two panels of Figure 2. The reader needs a map view equivalent to Figure 1 to get a feel for the glacier extent and geometry (for both glaciers it would be helpful to see a few contour lines which helps identifying steep parts and planes) and more explanations, otherwise it seems that Tinguiririca Glacier was half-heartedly added to the study.

*We agree that this extra example is difficult to integrate in the research context without providing further details. The available remote sensing data (high resolution images, and DEMs) for Tinguiririca was even scarcer than for the Leñas glacier, which did not allow for a fuller and more comprehensive description. We have taken the decision, based also on another referee's judgment, to discuss and compare the Tinguiririca event only very briefly in*

*terms of the avalanche volume and runout distance. This means that we have also eliminated the figures showing the Tinguiririca glacier.*

SURGE HISTORY The topic of glacier surging receives little attention in the manuscript. Do the satellite DEM's provide some hints for surge behavior? In any case, it would be good to write 1-2 sentences on this subject to put the collapse into context of the Aru Co and Kolka events. This could be built into the second paragraph of the Discussion section. Currently, there is some mentioning of a thermal regime change, but no specific evidence or context is provided.

*We thank the referee for the suggestion. We have added a few sentences about surges in the region. In a recent review of glacier surges in the Central Andes, Falaschi et al (2018, Progress in Physical Geography) found evidences of glacier surges at nearby glaciers, but not for the Leñas glacier specifically. We re-examined the material from Falaschi et al. and our own dataset and concluded that no evidence of a Leñas surge could be identified.*

*For clarification, the text referred to the overall thermal regime beneath the glacier but not specifically to the possibility of a thermally triggered Svalbard-type surge.*

MINOR COMMENTS Figure references: At several parts of the manuscript it is not clear what the autor's assertions are based on. For example, in the first paragraph of Section 3 no references to figures are made, but if I understand correctly the described observations are based on information shown in the figures.

*We agree that the figures could be better referenced in the main text and have inserted further links to them.*

Line 68: It would help to give a rough estimate of avalanche volumes for the 16 events of the WGMS.

*This is a good idea! Now included in the text.*

Line 134: "(e.g. due to decrease in glacier slope)" is unclear.

*We meant that the new debris cover might have developed in a now flatter glacier. We have clarified this in the text.*

Line 140-141: Do detachment scarp and crevasses really disappear or were they simply covered by debris?

*The scarp and crevasses were swept away in the avalanche, as they formed its head. No evidence of them being filled with debris was observed in the field.*

Near Line 150: How was the absence of bedrock beneath the glacier confirmed? Using boreholes? Could exposed bedrock be concealed by deposited sediments? Please mark/annotate figures accordingly.

*The absence of a hard bed beneath the glacier was visually evaluated in situ. No rock outcrops were to be found in the failed area whatsoever. The area is steep and subjected to rockfall, hence boreholes were not considered. From the thick sediment layer in the failed glacier area (visible in incised gullies seen in figures 3a and b) we believe the hard bed lies well beneath the glacier bed. We now mention this characteristic in the caption of Figure 3.*

Line 152: Reference to Figure 4a is unclear. Please mark/annotate figure accordingly.

*See previous comment. This has now been clarified (annotated).*

Line 156-157: The smoother hammocks and thermocarst should be highlighted in the respective figures.
*We have marked thermokarst ponds and hummocks in Figure 3a.*

Section 4: It would help to show parts of the meteorological analysis in a plot. Also, some specifics on the acceleration criteria would be of interest.
*We have chosen not to include a graph showing the meteorological data as this does not reveal a strong link with the collapse event. With 3 multipanel figures, we are already in the maximum length advised for a brief communication in TC journal.*

Line 183 (and elsewhere): It would help the non-expert to specify what is mean by "ordinary" ice avalanches.
*We agree and have changed this to "from steep fronts and hanging glaciers steeper than 30º.*

Discussion: It may be worth considering the possibility that the event happened as a series of small break-offs rather than a single rupture. Such cases are known to exist and it is not clear which conditions favor one scenario over the other. https://www.geopraevent.ch/project/weissmies-glacier-velocities/?lang=en
*Indeed, we can not be 100% sure if the event was due to a single rupture or several smaller ones. However, the field evidence and the relatively large blocks of massive ice point in the direction of an avalanche composed of large blocks (see Fig. 3b). We mention the possibility of small events now in the revised manuscript. We assume that numerical modelling could help in identifying this, but this is beyond the scope of the current study.*

Line 229 and following paragraph: When discussing the permafrost conditions it seems that the authors present arguments for and against permafrost. It was not clear to me what the actual conditions are believed to be. Also, it is not clear what the implication of the last sentence is (reference to Kolka).
*Confirming the presence/absence of permafrost in the plateau (i.e away from the headwall) would need proper, in-situ temperature logging. Alternatively, an approximation of the thermal conditions could be investigated by building a potential incoming solar radiation model. Again, we consider that any of them would be beyond the scope of the current study, and would not help in elucidating a collapse trigger per se. Due to the lack of convincing evidence for permafrost presence/absence in the pleateau where the ice avalanche deposit sits up to this point, we have removed the discussion on the role of permafrost in the preservation of the ice avalanche deposit.*

Line 244: Specify "same method".
*Checked and corrected. By 'same method' we referred to SRTM and ALOS PRISM differencing.*

Line 262: "time difference" → time lag.
*Changed to time lag.*

---

## Author Comment (AC3) · 15 Feb 2019

General comments: This study deals with the collapse of an unnamed glacier in the central Andes of Argentina in March 2007 (between 5 March and 14 March 2007). This glacier, named by the authors, ' Lenas glacier ', is located in the very remote area. The collapse and the avalanche have not been observed directly and it seems to remain unnoticed during several years. Very few data are available about this collapse. Most of data come from satellite images. This study aims at reconstructing the conditions of the collapse (Volume, slope, meteorological conditions, sismic events) in order to understand the possible causes of this breaking off. The authors claim that this event, very rare, can be compared with the very large collapses of Kolka in 2002 and Aru glaciers in 2016, given that the volume size is huge and the slope of the glacier is low. Unfortunately, the analysis and the conclusions remain qualitative and speculative due to the lack of data. This study contains vagueness, large assumptions, and lack of rigour for the following reasons:

*We thank the referee for the in-depth review. In regards to his/her main comments, we agree that the data available is limited and probable causes of the event are difficult to assess In our view, the chosen format of a Brief Communication, is perfectly suited to present and report this Leñas update and to make the community aware of the event. We cannot expect to get an answer for every open question from image time-series analysis and geomorphometric interpretation of surface characteristics (in the field and using DEMs / satellite data) for an event discovered with many years delay. The kind of event described in our study is rare enough that knowledge about every single event is important.*

*Also, brief communications do not normally require lengthy accounts of theoretical background and methodology principles, and thus we had not extensively elaborated on i.e. DEM accuracies and limitations of the DEM difference method, which can be nevertheless found in the literature.*

1) First, the uncertainty relative to the collapse volume can be questioned. The volume changes have been assessed from satellite images (Spot 5 12 February 2007, Landsat 5 March 2007, Spot 5 14 March 2007, Quickbird 19 April 2007). The failed glacier area and thickness have been estimated from images 12 February 2007 and 19 April 2007. The accuracy of each DEM is not mentioned. The authors wrote that ' the average thickness at the scarp was roughly 35 m as estimated from scarp shadows and solar angles at the time of acquisitions . Thus assuming a linear decrease of the glacier thickness from the scarp to the former glacier front . . .. . .a rough estimate of 4.5 106 m3 ' (l. 108-111) . No detailed information is given about the method and the uncertainty of thickness. In the following lines (l.112), the authors mentioned a ' conservative 15% error from uncertainty in detached area and thickness estimate' without providing any details about this uncertainty. Another ' independent ' estimate has been done from the difference between SRTM DEM (February 2000) and ALOS PRISM DEM AW3D obtained between 2007 and 2011. Again, the uncertainty of these DEM is not given. The uncertainty related to SRTM penetration is not mentioned. The authors mentioned only that the assumption of no radar penetration is confirmed by the comparison between C Band and X Band SRTM which show no significant difference.

The authors did not provide any detail or reference. In addition, the authors used the '
average DEMS' of ALOS PRISM DEM (2007-2011) with an ' average year' of 2009.
They assume that there is no change between March 2007 and 2009 (l. 125-126). It is
very confusing. The uncertainties relative to this assumption are not explained. Nothing
is said about the elevation differences between these 2007-2011 DEMs. The uncertainty
of 2.3 m (l. 128) on elevation difference seems to be very optimistic. Moreover, from
Figure 2a, one can see surprising elevation changes of 25- 50 m in several areas of the
upper part of the glacier between 2000 and 2009 far from the detached zone. These
values are similar to the elevation changes of the collapse area. However, no
explanation is given about that. Due to the lack of information, it seems very difficult to
assess the uncertainty on the volume of the collapse.

*We have now partially recalculated our volume estimates, and largely rewritten,
extended and rearranged the related description. We use now SRTM, ALOS
PRISM, and (new) the TanDEM-X DEMs as main source of our volume
estimate, and use the scarp height only as rough check. Note that we don't
need to estimate the DEM accuracies but only the accuracies of elevation
differences to arrive at an accuracy estimate for the volume. A detailed
assessment of the DEMs used is out of scope for our brief communication. The
gross uncertainties seen in the figure are situated on the steep headwall of the
glacier, whereas the detachment happened from the flatter part for which the
elevation differences to the left and right of the detachment are more
representative. Most importantly, we believe the exact number of the volume is
not crucial as we are only interested in the ballpark of the volume, i.e. around 4
Mm3, which we hope to demonstrate sufficiently now.*

2) The discussion about the mean slope' is confusing. The authors make a difference
between hanging glaciers with steep slopes (>30° ) and glaciers with low slope (lines
26-27). They wrote that ' the detachment of large portions of low-angle glaciers is much
less frequent ' (l. 34-35). The manuscript is confusing because the authors mentioned
both the low angle of reach (5° ) (lines 22 and 106), the average slope of glacier (24.6° )
(line 97) and the slope of the detachment part to discuss the stability/instability of the
glacier. These slopes are mentioned in different sections of the manuscript which
creates confusion.
The low angle reach is not relevant to study the stability of hanging part of the glacier.
More specifically, the ' suprisingly low angle of reach (5° ) ' (Abstract, l. 22) seems to
be irrelevant as an indicator of stability of glacier. The slope of ' detached glacier',
which seems to be the relevant value to assess the stability, is mentioned in Discussion
only in line 186 (15.6° ). We do not have any information about the method used to
calculate this slope. Is the surface slope before the collapse ? calculated on which
distance ? is the surface slope after the collapse ? Which images have been used to
obtain this value ? What is the accuracy of this calculated value ? The analysis of slope
change reveals also a lack of rigour. In line 142, the authors wrote : Ân´ the glacier
slope decreased from 24 .6 to 20.4° from before to after collapse Âz. The ˙ distance on
which this slope is calculated is not specified. One can assume that the slope change is
mainly due to the length changes of the glacier and the size of the avalanche. In this
way, is the slope change a relevant information?

*In order to clarify the different angles mentioned in the manuscript and avoid
confusion, we have now differentiated only 2 of them.*

*-The angle of reach is measured from the scarp head to the avalanche terminus. While this is not specifically relevant to the stability of the glacier, it does tell about the fahrboshung mobility index and we have therefore included it.*

*-The angle of the detached part of the glacier before collapse (15.6◦) was measured from the SRTM DEM over the failed glacier area measured from the Quickbird 2007 scene (mentioned in the text). SRTM has been tested in a large number of scientific studies in glacier areas and has been fully accepted to derive glacier topographic parameters (such as slope) with adequate results (e.g. Racoviteanu et al, 2009 in Annals of Glaciology, Frey and Paul, 2011 in International Journal of Applied Earth Observation and Geoinformation). Also, SRTM accuracies can be found in Farr et al. 2007 in reviews of Geophysics. We agree that the overall glacier slope change is not relevant information and have removed the previous analysis. Incidentally, as opposed to the referee's understanding, there was no hanging part of the glacier before collapse).*

3) As mentionned by the authors in Conclusions, due to the limited data, this study is not able to identify the causes of the Lenas event. Many assertions are highly speculative. For example ' the thin glacier front could have been frozen to the bed and a change in this polythermal regime may have caused changes in stability' (l. 202-204) or ' we suggest that the soft glacier bed material could have played an important role in the collapse. . ..' (l. 206-207), or ' we hypothesize a mixed origin for the debris layer observed on the ice avalanche deposit. . .' or ' . . .may indicate that a large glacier collapse has not happened in 2007 for the first time. This speculation relies on. . .. .. '. The Discussion is a list of assumptions and questions and does not shed light on the causes of this collapse.

*We agree that the discussion section includes speculative statements that partly go beyond the empirical evidence. Although we think this is allowed when speculations are clearly marked as such (e.g. to stimulate further research) we have removed most of them and focused on statements with at least some evidence.*

In summary, the authors claim that the Lenas collapse deviates from typical ice avalanches from steep glacier and can be compared to the rare low-angle glaciers collapses similar to Aru glaciers and Kolka glaciers avalanches.

*We agree that a direct comparison of the event observed here to the collapses at Kolka and Aru should not be made and have revised the text accordingly. However, we think when talking about detachments of glacier sections with comparably low surface angles, and about glacier avalanches travelling large distances over comparably flat surfaces it is appropriate to at least mention the Kolka and Aru collapses.*

Given the lack of information given in this study, the uncertainty on collapse volume can be questioned.

*We actually think that the derived collapse volume is a comparably robust part of the study and has higher certainty than several other numbers. We however revised the volume estimate parts significantly (see response to above comment).*

In addition, the data provided by this study are poor and do not allow to identify the possible causes of the collapse.

*We agree that the data available are limited and possible causes of the event are difficult to derive from it. However, we also think that sufficient information is around to make the community aware of the event. This is why we have chosen the format of Brief Communication, which is to our best understanding among others meant for such types of updates.*

This study points out the low detachment slope (15.6◦) although the determination of the slope has not been explained and the uncertainty on this slope is unknown. 'No significant change in glacier geometry could be identified due to the lack of data' as mentioned in Conclusions (l. 267-268).

*We have now added how the slope has been calculated and revised the text sections about the angles involved (see response to above comment).*

The authors suggest that soft bed characteristics play a crucial role on the collapse trigger. I do not think that this study provides sufficient quantitative information for understanding complex processes in glacier instabilities and collapses.

*We agree that there is very limited evidence for this speculation and briefly mention the idea in the discussion section.*

I do not think this study shed new light on the triggers and factors responsible for this event. Given the paucity of data, I am not sure that this event can be compared to Aru glaciers and Kolka glaciers avalanches as claimed by the authors.

*We agree that this direct comparison is based on very limited evidence and have rewritten this section (see above).*

---

## Editor Decision (ED1)

Dear authors

I am pleased with your revision. I agree with some general concerns raised by some of the reviewers that the contribution does not include new science, but I can truly see the value of reporting on this extreme even. Its description is state-of-the-art and I consider your contribution as suited for TC as short contribution.

Some minor technical comments:

Line 106 (and elsewhere): please consistently use elevation (instead of altitude).

Line 111: Suggest rewording. Difference?

Line 123: Please consistently add the multiplication sign: "×", i.e. $4.0 \times 10^6 \, \text{m}^3$

Lines 196-199: This sentence is a bit hard to follow. Suggest rewording. Maybe, … between steep hanging glaciers and the massive collapses…

Line 204: What is the source slope? Suggest rewording.

Line 236: fine grained deposits?

11 March 2019/Jürg Schweizer.